# Mass Spectrometry-Based Proteomic Technology and Its Application to Study Skeletal Muscle Cell Biology

**DOI:** 10.3390/cells12212560

**Published:** 2023-11-01

**Authors:** Paul Dowling, Dieter Swandulla, Kay Ohlendieck

**Affiliations:** 1Department of Biology, Maynooth University, National University of Ireland, W23 F2H6 Maynooth, Co. Kildare, Ireland; paul.dowling@mu.ie; 2Kathleen Lonsdale Institute for Human Health Research, Maynooth University, W23 F2H6 Maynooth, Co. Kildare, Ireland; 3Institute of Physiology, Faculty of Medicine, University of Bonn, D53115 Bonn, Germany; swandulla@uni-bonn.de

**Keywords:** mass spectrometry, muscle cell biology, muscle proteomics, myology, organelle proteomics

## Abstract

Voluntary striated muscles are characterized by a highly complex and dynamic proteome that efficiently adapts to changed physiological demands or alters considerably during pathophysiological dysfunction. The skeletal muscle proteome has been extensively studied in relation to myogenesis, fiber type specification, muscle transitions, the effects of physical exercise, disuse atrophy, neuromuscular disorders, muscle co-morbidities and sarcopenia of old age. Since muscle tissue accounts for approximately 40% of body mass in humans, alterations in the skeletal muscle proteome have considerable influence on whole-body physiology. This review outlines the main bioanalytical avenues taken in the proteomic characterization of skeletal muscle tissues, including top-down proteomics focusing on the characterization of intact proteoforms and their post-translational modifications, bottom-up proteomics, which is a peptide-centric method concerned with the large-scale detection of proteins in complex mixtures, and subproteomics that examines the protein composition of distinct subcellular fractions. Mass spectrometric studies over the last two decades have decisively improved our general cell biological understanding of protein diversity and the heterogeneous composition of individual myofibers in skeletal muscles. This detailed proteomic knowledge can now be integrated with findings from other omics-type methodologies to establish a systems biological view of skeletal muscle function.

## 1. Introduction

Modern cell biology is concerned with the detailed analysis of the structure and function of the basic cellular units of life and how they form highly complex structures in the form of cells, tissues, organs, bodily systems and individual organisms. This includes the detailed elucidation of protein expression levels, protein modifications and protein dynamics on the molecular level within the cellular context [1]. The application of mass spectrometry (MS)-based proteomics had a considerable impact on molecular cell biology [2], focusing especially on the analysis of cellular proteomes and organellar subproteomes [3]. The establishment of proteomics as both a cell biological concept, i.e., defining the entire dynamic set of proteins expressed by a given biological entity as a proteome [4], and a comprehensive bioanalytical approach, i.e., using large-scale protein separation techniques in combination with peptide/protein MS methods and bioinformatic software programs for high-throughput protein identification or the targeted characterization of isolated proteins [5], has revolutionized modern cell biology.

Of central importance for proteomic research is the biochemical definition of dynamic proteoforms being the basic units of a given proteome [6,7,8,9]. The proteolytic processing and post-translational modifications (PTMs) of individual protein species play a key role in their stability, biochemical functioning and subcellular localization [10]. Importantly, proteomics has reached single-cell resolution [11], giving unprecedented insights into the dynamic nature of the protein constituents that mediate cellular processes [12]. Major MS-based approaches in molecular cell biology include bottom-up proteomics and top-down proteomics [13]. Bottom-up proteomics aims at the peptide-based analysis of large and diverse protein populations [14,15], while top-down proteomics has a proteoform-centric focus and is concerned with the biochemical identification and detailed characterization of purified and intact protein species [16,17]. The bioanalytical combination of both methodologies can be strategically employed for improved data integration [18,19]. The skeletal muscle proteome project [20] was initiated in 2001 using initially top-down and gel-based separation methodology [21,22]. Subsequently, MS-based surveys have been widely applied to study almost all crucial aspects of skeletal muscle cell biology using both bottom-up and top-down proteomics [23,24]. This has led to the creation of a detailed proteomic map of different types of skeletal muscles [25,26,27,28] including predominantly fast-twitching versus slow-twitching muscles [29,30,31,32] and the exhaustive biochemical profiling of individual contractile fiber types [33,34,35,36].

A large number of extensive reviews have previously been published that cover specific biochemical and cell biological approaches and the underlying scientific scope of proteomic profiling with a focus on voluntary muscle tissues. These articles give a comprehensive overview of the many applications of MS-based analyses to improve our molecular and cellular understanding of skeletal muscle protein diversity, modifications and dynamics. Previous reviews have extensively covered the proteomic analysis of human skeletal muscle [20,37], the impact of exercise [38,39,40], disuse-related muscular atrophy [41], skeletal muscle plasticity [42,43], neuromuscular diseases [44,45] and frailty-associated muscle aging [46,47]. Building on these articles, we provide here a general synopsis of skeletal muscle proteomics and its scientific importance for the field of basic and applied myology. Examples of skeletal muscle proteomics are discussed, as well as an outline given on the impact of genomic, post-transcriptional and post-translational factors on muscle proteome diversity.

## 2. Mass Spectrometry-Based Proteomics to Study Skeletal Muscle Cell Biology

Expression changes, biochemical modifications and dynamic adaptations of proteins in their cellular context are of critical importance for understanding the molecular pathways that underlie fundamental biological processes. This positions MS-based proteomics at the core of the comprehensive bioanalysis of cell biological mechanisms, such as the excitation-contraction-relaxation cycle in skeletal muscles.

### 2.1. The Importance of Proteomics and the Concept of Proteoforms in the Biological Sciences

Since the initiation of proteome-wide research initiatives in the 1990s [48], MS techniques were widely applied for the comprehensive analysis of a variety of cells, tissues and organisms, including the human proteome [49,50,51,52], which had enormous consequences for the scientific scope of biological and biomedical research [53,54,55]. At the level of individual protein species, the term proteoform [6,7] best describes the basic unit of the proteome [56,57] and the central question remains of how many proteoforms exist within a given proteome [58,59,60]. Genomic variations at the level of allelic or locus heterogeneity, combined with post-transcriptional alternative RNA splicing events and extensive PTMs of proteins result in a highly complex and dynamic proteoform expression pattern [10]. Therefore, most single genes appear to be associated with a family of proteoforms rather than a sole protein product. The approximately 20,000 human genes are estimated to produce millions of distinct protein species [61]. This gives proteoform-focused research a central space in modern cell biology and promises to shed light on how protein heterogeneity is mediated by transcriptional/translational modifications in combination with the formation of higher-order protein structures and dynamic protein–protein interactions resulting in distinct biological phenotypes [62].

The rapid developments of novel proteomic technologies and applications has led to the establishment of many specialized approaches focusing on targeted biomarker proteomics [63,64], clinical proteomics [65,66], drug discovery and pharmaco-proteomics [67,68,69], proteo-genomics [70,71], membrane proteomics [72,73], extracellular matrix proteomics [74,75], native proteomics [76,77,78], thermal proteome profiling [79], the analysis of protein conformations and interactions via chemical cross-linking MS [80,81] and subcellular/organelle proteomics [82,83]. The more recently developed discipline of native MS focuses on the detailed top-down characterization of structural heterogeneity in protein isoforms under non-denaturing conditions, making this approach an important part of structural cell biology [84,85]. Together with genomics, transcriptomics, metabolomics, lipidomics, glycomics and cytomics, modern proteomics is embedded within a multi-omics research environment and plays an integral part in systems biology [86]. Techniques beyond MS include antibody-based screening methods for establishing a comprehensive protein expression atlas in all major tissue types [87] and next-generation proteomics [88] leading to a more comprehensive view of biological function through integromics research [89].

### 2.2. Large-Scale Protein Separation and Top-Down Proteomics

In the field of skeletal muscle proteomics, the isolation of native protein arrangements and intact proteoforms is of crucial importance for the detailed analysis of muscle protein function [23,24]. The initial protein separation and isolation from complex samples prior to MS analysis is usually carried out using subcellular fractionation procedures [90] involving (i) differential centrifugation or gradient ultra-centrifugation [91], (ii) affinity agglutination of distinct membranes [92], (iii) isoelectric focusing [93], (iv) one-dimensional gel electrophoresis [94], (v) two-dimensional gel electrophoresis [95] and/or (vi) reversed-phase liquid chromatography [96].

For gel-based top-down proteomics, which is routinely used for the isolation of intact proteoforms prior to controlled proteolysis and peptide MS analysis [6,97], the optimum application of two-dimensional gel electrophoresis (2D-GE) is of central importance [98,99,100]. Using isoelectric focusing (IEF) in the first dimension and sodium dodecyl sulphate-polyacrylamide slab gel electrophoresis (SDS-PAGE) in the second dimension [101,102,103], proteins can be efficiently separated by the unique combination of their overall charge and molecular size [104,105,106]. Recently, micro-needling was introduced to improve the time requirements for the IEF step in 2D-GE [107]. Proteins can be separated under both denatured or native conditions. In addition to using sodium dodecyl sulfate (SDS) exclusively in second dimension gels [108,109,110], 2D-gels can be operated with combinations of alternative detergents to improve the resolution of hydrophobic membrane proteins [111]. A specific method uses in the first dimension the cationic detergent benzyldime-thyl-n-hexadecylammonium chloride (BAC) and in the second dimension the anionic component SDS; therefore, this technique is abbreviated as BAC/SDS-PAGE [112]. Two-dimensional blue native polyacrylamide gel electrophoresis (BN-PAGE) employs native conditions [113,114] and is especially useful for studying supramolecular protein complexes present in mitochondria [115]. Diagonal non-reducing/reducing two-dimensional gel electrophoresis can be used to exploit both natural or modified differences between proteins and protein complexes during large-scale separation approaches following chemical cross-linking [116,117,118]. Suitable and highly sensitive dyes have to be used for the optimum visualization of two-dimensionally separated protein spots and subsequent sample picking for MS-based analysis [119,120,121].

Labelling of different proteoforms can be achieved using differential fluorescent tagging with CyDye2, CyDye3 and CyDye5 agents prior to 2D-GE [122,123]. High-throughput comparative studies can be performed via comparative two-dimensional fluorescence gel electrophoresis (CoFGE) [124,125]. Difference gel electrophoresis (2D-DIGE) can be carried out with 2-CyDye or 3-CyDye systems using saturation versus minimal labelling approaches [126,127]. Of note, 2D-GE, including the 2D-DIGE technique, can be employed effectively to study native protein configurations [128,129], as well as PTMs using specific chemical stains or blot overlay methods. Phosphoprotein gel stains, e.g., Pro-Q Diamond, provide a useful approach for selectively staining phosphoproteins in 2D gels [130]. By combining a standard DIGE or SYPRO Ruby experimental workflow with phospho-specific fluorescent Pro-Q Diamond staining [131], the sensitive and quantitative detection of phosphoproteins in 2D gels can be accomplished [132,133]. Other stains or blotting techniques routinely used include those targeting glycosylated proteins in 2D gels, e.g., Pro-Q Emerald [134] or peroxidase-conjugated wheat germ agglutinin [135], providing a rapid and very sensitive methodology for detecting glycoproteins of interest [136]. A more comprehensive description of PTM analyses is provided below. As an alternative to 2D-GE, multi-dimensional protein identification technology (MudPIT) can be employed, which is a non-gel technique for protein separation that uses two-dimensional liquid chromatography prior to MS-based analysis [137,138]. The initial preparative steps involved in the characterization of skeletal muscle proteins using top-down proteomics versus bottom-up proteomics are summarized in Figure 1.

Very large protein species that can often not be properly separated by conventional 2D-GE can be analyzed by combining 1D-GE, liquid chromatography and tandem MS (GeLC-MS/MS) methodology [139,140,141]. The stabilization of sensitive protein–protein interactions in skeletal muscle tissues can be supported by chemical cross-linking [142], a method that is becoming increasingly used to study dynamic protein conformations [80,81,84]. Especially the quantification and structural analysis of integral proteins [143] using MS-based methods is a challenging task [144,145,146]. High-resolution native MS techniques have been developed to study the dynamic nature of protein complexes [147,148,149].

### 2.3. Sample Preparation, Protein Digestion and Bottom-Up Proteomics

For bottom-up proteomics, a variety of sample preparation approaches have been developed for optimum protein extraction [150,151,152], including the widely employed filter-aided sample preparation (FASP) method that can be used to exchange buffers and remove incompatible detergents prior to MS-based analysis [153]. The routine bottom-up proteomic analysis pipeline can be divided into a series of consecutive steps [152] comprising (i) sample harvesting from complex cellular mixtures, (ii) sample pre-treatment, such as enrichment procedures using differential centrifugation or affinity isolation of protein constellations, or depletion procedures to enrich low-abundance proteins, (iii) initial sample protection from excess proteolysis using suitable protease inhibitor cocktails, (iv) efficient homogenization using mechanical grinding approaches or sample pulverization with liquid nitrogen, (v) protein extraction via precipitation or other biochemical methods, (vi) protein quantification using sensitive dye binding or other reliable assay systems, (vii) chemical reduction and alkylation of protein samples, (viii) protein digestion using enzymes such as trypsin for the generation of distinct peptide populations, (ix) peptide fractionation and MS analysis, (x) protein identification, and (xi) systems bioinformatic analysis of identified proteoforms and their PTMs [154,155,156]. Figure 2 compares the main differences in the preparation of protein samples between top-down proteomics and bottom-up proteomics, i.e., the isolation of intact proteoforms versus the enrichment of protein fractions.

Specific requirements of the starting material, i.e., cells, tissue specimens or biofluids, have to be taken into account for the most suitable extraction of protein populations. For the detection of muscle-derived marker proteins in biological fluids such as serum, saliva or urine, biofluids should be screened with optimized proteomics platforms [157] and usually require immuno-depletion, immuno-enrichment or offline fractionation techniques for the optimum isolation of low-abundance proteins [158,159,160]. In contrast, unfractionated and crude tissue samples can be conveniently prepared using FASP [161], In-StageTip (iST) [162], suspension trapping (S-Trap) [163], single-pot solid-phase-enhanced sample preparation (SP3) [164] or universal solid-phase protein preparation (USP3) [165] methodology. Importantly, the application of single-cell isolation is becoming more crucial for sample preparation in proteomics [166]. For the dependable screening of thousands of prioritized peptides with increased proteome depth, the prioritized Single-Cell ProtEomics (pSCoPE) analysis platform was developed [167]. Some of the bioanalytical issues and technical limitations encountered with the lysis and proteolytic digestion of cell or tissue samples [168] can be overcome with pressure cycling technology (PCT) [169].

Following isolation, the controlled digestion of proteins can be performed via in-solution [170,171], in-gel [172,173] or on-membrane [94,174,175] treatment with a suitable enzyme such as trypsin [176] or a great variety of other proteases used alone or in combination, including chymotrypsin, Glu-C, Asp-N, Arg-C, Lys-N and Lys-C [177,178,179]. Recently, a more rapid in-gel digestion protocol was developed to improve the GeLC-MS/MS technique [180], which can be applied to study very large proteins that cannot be separated by conventional 2D-GE [139]. Using BAC cross-linked gels, these separation media can be rapidly solubilized via chemical reduction. This new method has been named BAC-gel dissolution to digest PAGE-resolved objective proteins (BAC-DROP) [181].

### 2.4. Mass Spectrometric Protein Identification

Routine protein identification can be carried out using matrix-assisted laser desorption/ionization time-of-flight (MALDI-ToF) MS or electrospray ionization (ESI) MS analysis [182]. Liquid chromatography and the fragmentation of peptides can be employed to determine the amino acid sequence and PTMs of interest via tandem MS analysis (MS/MS) [183,184,185] and a variety of related MS methods [186,187], including single-cell MS [188]. A ToF-based analyzer determines efficiently the mass-to-charge ratio (m/z) using the time ions require to travel through an electric field under vacuum [189]. This is combined with the MALDI-mediated detection of peptides [190]. The optimized usage of reversed-phase liquid chromatography is crucial prior to the direct injection of peptides using ESI [191] or related techniques for MS analysis [192]. The detailed characterization of proteoforms can be carried out using Fourier-transform ion cyclotron resonance MS (FTMS) [193]. MS analyses can be performed with label-free quantification (LFQ) approaches [194,195] or quantitative label-based methods [196,197].

The acquisition of data sets in MS-based studies can be attained via data-dependent acquisition (DDA) [198], data-independent acquisition (DIA) [199] or targeted data acquisition (TDA) [200]. DIA-based methods are ideal for bottom-up proteomics focusing on the large-scale and LFQ analysis of proteoforms [201]. In the specialized biochemical discipline of targeted proteomics [202], multiplexed protein quantification can be carried out with the help of selected/multiple reaction monitoring (SRM/MRM) [203] and parallel reaction monitoring (PRM) techniques [204], which are characterized by high degrees of bioanalytical accuracy, specificity and reproducibility [205]. Targeted label-free data-acquisition can be performed with Sequential Window Acquisition of all Theoretical Mass Spectra (SWATH) [206,207]. Quantitative label-based methods used in MS-based analyses include stable isotope labelling using amino acids in cell culture (SILAC) [208,209], the isobaric tagging for relative and absolute quantitation (iTRAQ) technique [196,210], isotope-coded affinity tags (ICAT) [211,212], isotope-coded protein labelling (ICPL) [213] and isobaric tandem mass tagging (TMT) [197,214].

Advances in instrumentation and proteomic screening methodology are represented by BOXCAR, RTS-SPS-MS3, FAIMS, Evosep, TIMS, SOMAscan and PEA. BOXCAR is a data-independent acquisition method that fills multiple narrow mass-to-charge segments and allows for in-depth proteomic profiling [215]. RTS-SPS-MS3 is based on real-time search (RTS) synchronous precursor selection (SPS)-assisted acquisition, which improves proteomic coverage especially in multiplexed single-cell analyses [216]. FAIMS stands for high-field asymmetric ion mobility spectrometry. This method uses a mass spectrometer that is equipped with a front-end FAIMS interface that improves proteomic coverage [217]. Evosep is a liquid chromatography system for improved high-throughput peptide analysis [218]. TIMS uses trapped ion mobility spectrometry with parallel accumulation serial fragmentation (PASEF) operation mode and provides high analytical speed and sensitivity [219]. In addition, multiplexed proteomic assays have been developed for the parallel assessment of large numbers of proteins in biofluids using both modified aptamers (SOMAscan) and proximity extension assays (PEA) for alternative immuno-analysis [220,221]. Major methods used in the MS-based analysis of skeletal muscles and proteomic data acquisition are summarized in Figure 3.

Verification analyses to confirm the MS-based identification of proteomic changes on the biochemical and cellular level can be conveniently performed using standardized immunoblotting [222,223], enzyme assays [224,225], confocal microscopy [226] and histochemical analyses [227].

### 2.5. Microproteomic Analysis Using Laser Capture Microdissection

A key technique of molecular cell biology, laser-assisted microdissection [228], can be conveniently adapted for microproteomic applications [229] and used to study protein populations in small cellular regions of interest [230,231]. In general, the highly focused isolation of cellular structures employing direct microscopical visualization of tissues can be used to generate detailed systems biological maps through the bioanalysis of pure histological specimens [232]. Within a targeted microscopical area, the application of laser-assisted microdissection can be employed to determine spatial signatures of distinct cellular layers. Micro-dissected tissue specimens can be used to generate cDNA libraries, the systematic genotyping of DNA, RNA transcript profiling, detailed protein pathway analyses, advanced biomarker discovery and spatial proteomics [233]. Optimized microproteomic approaches have been instrumental for advanced MS-based analysis of cell type-specific properties and their cell biological microenvironment [234].

To avoid analyte loss during sample preparation for spatial proteomics, cellular specimens isolated via laser capture microdissection, or alternatively flow cytometry, should be properly prepared for the subsequent MS-based analysis. This can include carefully executed cell biological disruption methods, such as cellular lysis, protein extraction and sample clean-up, and optimized protein separation using liquid chromatographic or gel electrophoretic methodology [235]. Laser capture proteomics has been widely applied to carry out spatial analyses of diverse tissue types, including various skeletal muscle preparations [33,236] focusing, for example, on the analysis of myoferlin regulation during myofiber injury and regeneration [237], the determination of myosin content in individual human myofibers [238] and the proteomic assessment of the myotendinous junction [239].

### 2.6. Proteomic Analysis of Post-Translational Modifications

As already briefly outlined in the above section on the most commonly used MS approaches, proteomics is not only employed for the routine identification of proteoforms and the determination of abundance changes in individual proteins, but also represents an excellent method for the detailed analysis of PTMs [154,155]. Of note, major types of PTMs, such as phosphorylation, glycosylation, ubiquitination and methylation are essential for regulating protein stability, interactions and cellular localization [240]. Some PTMs can be added or removed to adapt and control the level of intracellular signaling cascade activation through the relationship between modifying and demodifying enzymes [241]. These enzymes include kinases, phosphatases, peroxidases, transferases, ligases and deubiquitylases, which can add or remove functional groups, with an estimated 5% of the proteome covering such enzymes. PTMs are increasingly investigated for their crucial role in physiology and pathophysiology [242]. Importantly, the detailed and specific detection of abnormal PTM events provides a potential reservoir of biomarkers for clinical utility, including early diagnosis, prognosis and to predict responses to therapeutic regimens in neuromuscular disease. A greater understanding of abnormal PTMs in specific proteins also provides an opportunity to accurately pursue such manifestations therapeutically and add to the already expanding list of drug targets in skeletal muscles. Accurate detection and analysis of PTMs is crucial for understanding their functional implications, and while technical difficulties limit these studies, recent advancements in these areas are now facilitating researchers in establishing a greater understanding of this complex area.

MS-based analysis of protein PTMs represents a sensitive and high-resolution approach that typically utilizes bottom-up proteomic approaches. The analytical approach relies on the controlled digestion of proteins for generating peptides using enzymes such as trypsin and/or lysyl endopeptidase (Lys-C) to perform this preparative action [154], as already outlined above. The enrichment of specifically modified peptides is an important consideration, as the creation of a pool of modified peptides, distinct from their unmodified counterparts, will reduce sample density and facilitate efficient PTM identification and quantification via MS. Peptide enrichment can be achieved using affinity strategies, targeting specific PTMs and concentrating their abundance in precise fractions. Affinity strategies can involve chromatographic separation based on unique chemical properties or purification based on antibody/protein domain recognition.

Immobilized metal affinity chromatography (IMAC) has been routinely used for phospho-peptide enrichment prior to MS analysis [243], with the use of positively charged metal ions including titanium (Ti^4+^), gallium (Ga^3+^) and iron (Fe^3+^) being central to this process [244]. Anti-phospho-tyrosine (pTyr) antibodies are frequently used to enrich tyrosine-phosphorylated peptides, as pTyr is significantly lower in abundance compared to phospho-serine (pSer) and phospho-threonine (pThr) [245]. Within advanced MS instrumentation, fragmentation of enriched peptides using approaches such as collision-induced dissociation (CID) or high-energy collision dissociation (HCD), can facilitate PTM identification. Phosphorylated peptides produce a characteristic neutral loss of the phosphate group, a prime example of this proteomic approach [246]. A similar antibody-based enrichment strategy can be used for the study of protein ubiquitination in human skeletal muscles, by utilizing antibodies that recognize a two glycine (GG) remnant on ubiquitinated peptides [247]. An overview of routine approaches used for the PTM analysis of skeletal muscle proteins is shown in Figure 4.

The quantification of PTMs can be performed using label-free approaches or label-based methods, with SILAC and TMT being routinely used. The phospho-proteomic analysis of human skeletal muscle quantified 8511 unique phosphorylation sites, with approximately 12% regulated due to physical exercise using iTRAQ or TMT combined with phospho-peptide enrichment [248]. A recent application of PTM analysis of human skeletal muscle proteins has focused on the modulatory effects of high-intensity interval training on contractile fibers [249]. The proteomic analysis of *vastus lateralis* muscle biopsies collected before and after training revealed distinct changes in the lysine-acetylome, an important PTM with respect to regulating process such as muscle metabolism, myofiber growth and muscle contraction [249].

For lower throughput PTM analyses, as compared to relatively swift MS-based approaches, immunoblotting is an appropriate technique facilitating the identification and quantification of PTMs by employing antibodies that specifically recognize modified protein targets. While this experimental technique is well validated, the generation of high-quality antibodies to PTMs is still demanding [250]. Protein sample probing using antibodies that are specific for both the modified antigen and its unmodified counterpart allows the level of PTMs to be evaluated for comparing specific sets of specimens. For example, phosphorylation in human skeletal muscle was successfully determined using antibody screening to study the effects of resistance exercise with a focus on type II myofibers [251]. The phosphorylation statuses of AMP-activated kinase (AMPK), eukaryotic initiation factor 4E-binding protein (4E-BP1), p70/p85-S6 protein kinase (S6K1) and ribosomal S6 protein (S6) could be determined using phospho-specific versus pan antibodies when comparing skeletal muscle biopsies before and immediately after exercise [251]. The pan class of antibodies recognizes a protein in its phosphorylated or unphosphorylated state, since it binds to an epitope that is distinct from the dynamically phosphorylated site. The availability of both phosphorylation-specific and pan antibodies can be exploited in enzyme-linked immunosorbent assays (ELISAs). Sandwich-based ELISAs employ two antibodies to detect phosphorylated proteins, with one antibody-binding total protein while the second antibody recognizes the phosphorylation site specifically. The results from these type of ELISAs can be quantitative when the phosphorylated protein of interest is available in purified form for standard curve generation. Changes in protein phosphorylation can then be used to investigate pre-selected proteoforms of physiological or pathophysiological importance. Notably, multiplex detection of phosphorylated proteins is feasible using sandwich-based antibody arrays. This methodological approach has recently been applied to study skeletal muscle wasting and protein turnover in pancreatic cancer cachexia [252].

## 3. Skeletal Muscles—The Proteomics Perspective

Muscle tissues account for approximately 40% of body mass, making contractile fibers one of the most abundant cell types in humans. Thus, cellular adaptations, biochemical changes, metabolic fluctuations or dysfunction of muscle fibers can exert a considerable influence on whole-body physiology [253].

### 3.1. Basic and Applied Myology

The human body contains approximately 650 individual skeletal muscles [254], which are involved in a large number of diverse biological functions, including (i) excitation-contraction-relaxation cycles to enable coordinated locomotion for body movements, (ii) postural maintenance of supportive structures to control muscle-skeletal balance and bodily protection, (iii) verbal and facial communication via the oral and facial musculature, (iv) respiration with the help of the diaphragm muscle, (v) regulation of body thermogenesis, (vi) provision of a protein reservoir for periods of starvation and (vii) metabolic and bioenergetic integration [255]. Figure 5 gives an overview of the extent of proteome-wide adaptations due to changed functional demands or pathophysiological alterations in the skeletal musculature.

The skeletal musculature is a highly adaptable organ system [42] that can be affected by changes in lifestyle, physical activity levels, side effects seen in traumatic injury, extensive pharmacotherapy and the natural aging process [38,46]. Acute or chronic myofiber degradation, inflammation and/or myofibrosis are involved in many primary muscle-associated diseases, such as muscular dystrophies [44,256], but also occur in association with various common pathologies, such as cancer, diabetes, obesity, heart disease or sepsis. Striking cellular changes in the skeletal musculature are observed during embryonic development, regenerative adult myogenesis, in response to different forms of physical exercise, disuse-related muscular atrophy, muscle fiber type specification, muscle transitions, primary neuromuscular disorders, muscle co-morbidities and sarcopenia of old age. Thus, a detailed understanding of normal muscle function and its cellular regulation including the dynamics of the skeletal muscle proteome, as compared to muscle wasting, is important for diverse biological and biomedical disciplines. This includes crucial aspects of lifestyle biology, health science and clinical medicine, including sports and exercise physiology [39,40], electro-stimulation therapy, research into skeletal muscle plasticity [43,257], disuse-related muscular atrophy and the effects of re-innervation [41,258,259], space flight medicine [260,261], neuromuscular diseases [44,45], frailty-associated muscle aging and biogerontology [47,262,263,264], forensic medicine [265,266] and the meat industry [267,268]. Important topics in muscle cell biology that are gaining steady interest are the regulation of skeletal muscle mass [269], myokine signaling [270] and muscle-bone-fat crosstalk [271]. Figure 6 summarizes important cell biological characteristics of skeletal muscles, including histological, physiological and biochemical properties of the two main types of contractile fibers, i.e., slow-twitching type I myofibers and predominantly fast-twitching type II myofibers [253,254].

In the adult human skeletal musculature, a mixture of slow oxidative type I, fast oxidative-glycolytic type IIa and fast glycolytic type IIx/d myofibers, plus hybrid fibers ranging from I/IIa to IIa/IIx, are found in most individual muscles [238,272]. Interestingly, in small rodents, the fiber type distribution patterns are more complex as compared to human voluntary muscles. Rodent muscles contain additional and extremely fast-twitching myofibers of the type IIb [273]. Excellent markers of the various subtypes of myofibers are represented by slow myosin heavy chain MyHC-1(beta) and the fast myosin heavy chains MyHC-2a, MyHC-2x and MyHC-2b [274]. The biochemical profiling of the contractile apparatus of skeletal muscles has recently been reviewed and this article lists the most abundant fiber type-specific markers, i.e., isoforms of myosin heavy chains, myosin light chains, troponins and tropomyosins, that were confirmed by proteomics [47]. The identification of individual fiber types using contractile apparatus proteins can be conveniently combined with reliable markers that belong to the Ca^2+^-handling complexes of skeletal muscles. This includes the slow SERCA2 and fast SERCA1 isoforms of the sarcoplasmic reticulum Ca^2+^-ATPase, and the slow CSQ2 and fast CSQ1 isoforms of the luminal Ca^2+^-binding protein calsequestrin (Figure 7).

The traditional determination of fiber type specification using relatively crude histological, histochemical or immuno-histological labelling techniques [226,227,275,276] has been superseded by high-throughput proteomic procedures that now focus on the biochemical profiling of distinct myosin proteoforms in single myofibers. This advanced approach for fiber type screening employs a proteomic analysis platform with a capillary liquid chromatography-MS gradient using a 96-well format [277]. This superb improvement of myofiber screening, named Proteomics-high-throughput-Fiber-Typing (ProFiT), shows the enormous impact MS has had already on skeletal muscle cell biology.

### 3.2. Proteomic Complexity in Skeletal Muscles

Within the various layers of biological organization in multi-cellular organisms, the relationship and flow of information from gene to RNA/mRNA to protein/proteoform to the cytome and functional physiome is highly complex, as outlined in Figure 8 for skeletal muscles. The underlying mechanisms that produce a considerably higher number of proteins per individual gene from the approximately 22,000 genes in the human genome are due to alternative promoter usage and alternative post-transcriptional RNA splicing mechanisms, as well as dynamic PTMs such as proteolytic cleavage of a protein product [54,55]. The combined involvedness of variations at the level of DNA, RNA and protein modifications are the underlying evolutionary drive that has produced a human proteome of astonishing complexity [278,279]. Especially high levels of alternative splicing contributes to the generation of proteomic diversity [280], which is also clearly the case in skeletal muscle tissues [281]. MS-based analysis can be instrumental to evaluate this considerable influence of RNA splicing on protein expression level in skeletal muscle fibers [23,24]. In humans, approximately 90% of transcripts of protein-coding genes have been estimated to undergo alternative splicing [278].

Some genes that contain protein-coding sequences for the production of muscle-specific components, such as the extremely large X-chromosomal *DMD* gene that encodes the Dp427-M isoform of the membrane cytoskeletal protein named dystrophin, have more than one promoter [282] (Figure 9). Thus, several protein products with tissue-specific expression patterns can derive from the sequence information that is stored in one individual gene. The production of precursor messenger RNA and alternative RNA splicing play a crucial part in the post-transcriptional regulation of gene expression. For example, the sarcoplasmic reticulum Ca^2+^-ATPase exists in the form of several protein isoforms with developmental and fiber type-specific expression patterns which derive through alternative RNA splicing from two genes [283] (Figure 9). Extensive PTMs are key mechanisms that produce distinct proteoforms, the functional protein units of cellular biological processes. An example is the dystroglycan complex which is part of the larger dystrophin complexome at the muscle fiber periphery [284]. The alpha/beta-dystroglycan complex is produced through proteolytic processing from one protein product that is encoded by the *DAG1* gene [285,286]. The two protein products are biochemically very different protein species; one is a typical transmembrane-spanning glycoprotein of the sarcolemma, beta-dystroglycan, and the other is an extracellular laminin receptor located in the basal lamina, alpha-dystroglycan (Figure 9).

### 3.3. Proteomic Profiling of Skeletal Muscles

The main bioanalytical avenues taken in the proteomic characterization of skeletal muscles include top-down proteomics focusing on the characterization of intact proteoforms, bottom-up proteomics, which is a peptide-centric method concerned with the large-scale detection of proteins in complex mixtures, and subproteomics that examines the protein composition of distinct subcellular fractions [20,37]. MS-based studies have decisively improved our general cell biological understanding of protein diversity and the heterogeneous composition of individual fibers in skeletal muscles [36]. This detailed proteomic knowledge that was accumulated over the last two decades can now be integrated with findings from other omics-type methodologies, such as genomics, transcriptomics, lipidomics, metabolomics and cytomics, to establish a systems biological view of skeletal muscle function. Figure 10 gives an overview of the bioanalytically accessible parts of the highly dynamic skeletal muscle proteome and the main subcellular structures of myofibers.

#### 3.3.1. The Status Quo of the Skeletal Muscle Proteome

Detailed information on the MS-based coverage of individual muscle proteins can be retrieved from multi-consensus files, which are usually provided as supplemental tables in the above- and below-quoted papers of major proteomic studies with a focus on skeletal muscles, or can be accessed from proteomic depositories and international databanks, such as ProteomeXchange (PX) [287] (http://www.proteomexchange.org). Currently, these detailed listings of MS-generated hits sum up to over 10,000 individual proteins that form the core proteome of skeletal muscles.

The continued expansion of the total number of identified muscle-associated proteins and the increased extent of established proteoform diversity has led to detailed proteomic maps of different types of skeletal muscles [25,26,27,28,288,289,290], including predominantly fast-twitching versus slow-twitching muscles [29,30,31,291,292,293,294]. Reliable fiber type-specific markers include fast versus slow isoforms of contractile proteins, such as myosin heavy chains, myosin light chains, troponins and tropomyosins, and Ca^2+^-regulatory proteins, including Ca^2+^-pumps, Ca^2+^-binding proteins and Ca^2+^-channels.

Established proteomic markers of adult fiber types [33,34,35,36] are:Myosin-7 (MyHC-I) for slow type I myofibers;Myosin-2 (MyHC-IIa) for fast type IIa myofibers;Myosin-1 (MyHC-IIx/d) for fast type IIx/d myofibers;Myosin-4 (MyHC-IIb) for extremely fast type IIb myofibers.

Frequently used markers of the contractile apparatus and its auxiliary filaments within the sarcomeric structure [25,26,27,290] include:Myosin light chain MLC1/3 for the thick myosin-containing filament;Alpha-actin ACTA for the thin actin-containing filament;Tropomyosin isoform TPM2 for the tropomyosin complex;Troponin subunit TnC for the troponin complex;Titin for the sarcomere-spanning titin filament;Myomesin MYOM-1 for the sarcomeric M-band;Alpha-actinin ACTN2 for the Z-disk complex.

#### 3.3.2. The Proteome of Specialized Cells and Structures within Skeletal Muscles

The proteome of intrafusal muscle spindles, which run anatomically in parallel to bulk extrafusal myofibers, are part of the proprioceptive system of the body that is intrinsically involved in controlling skeletal muscle length, coordinated movements and posture, was recently analyzed [295]. As compared to skeletal muscle fibers, the proprioceptive muscle system was shown to express higher levels of myosins-3/6/7/7b, collagen isoform COL-IV and sensory neuron proteins PIEZO2 and SLC17A7. Proteomic profiling was also employed to study the detailed molecular architecture of the neuromuscular junction with markers such as the nicotinic acetylcholine receptor, agrin, utrophin and the enzyme acetylcholinesterase [296,297]. Of crucial importance for the swift regeneration of damaged muscle fibers are myogenic stem cells, which are represented by the satellite cell population that is located between the sarcolemma and basal lamina in adult skeletal muscles. Satellite cell and activated muscle precursor cell proteomics has recently been carried out to identify biomarkers of fiber damage [298] and muscle aging [299].

#### 3.3.3. The Subproteome of Skeletal Muscles

In addition to cataloguing surveys of total skeletal muscle extracts, subcellular studies have identified a large number of organellar-specific proteoforms in skeletal muscles [300,301]. Subproteomics is an important subdiscipline of protein biochemistry that focuses on the identification and characterization of proteins that specifically associate with particular organelles using MS-based proteomics [82,302,303]. Subproteomic studies of skeletal muscle preparations were directed towards the surveying of proteins that are highly enriched in the sarcolemma [92], sarcoplasmic reticulum [175,304], mitochondrion-sarcoplasmic reticulum linker complexes [305], mitochondria [306,307,308] and the sarcosol [309], as well as the class of giant muscle proteins, such as dystrophin, nebulin, obscurin, plectin, the ryanodine receptor and the half-sarcomere spanning filamentous component titin [310]. Of note, the more recent application of single-cell proteomics (SCP) for studying fast versus slow myofibers [36] has revealed further molecular features of the extensive complexity of fiber type-specific protein expression patterns [33,34,35,261,277,311,312,313].

Robust skeletal muscle markers that are closely associated with distinct subcellular structures [25,92,304,305,309] include:Glycolytic enzymes for the sarcosol;Dystroglycan beta-DG for the sarcolemma;Caveolin-1 for surface caveolae structures;Dysferlin for the sarcolemmal repair apparatus;Talin for costameres involved in lateral force transmission;Laminin for the basal lamina;Collagen isoforms for the various layers of the extracellular matrix, including the epimysium, endomysium and perimysium;Dystrophin isoform Dp427-M for the sub-plasmalemmal membrane cytoskeleton;Alpha-1S subunit of the L-type Ca^2+^- channel for the transverse tubules;SERCA-type Ca^2+^-ATPases for the longitudinal tubules of the sarcoplasmic reticulum;Calsequestrin for the terminal cisternae region of the luminal sarcoplasmic reticulum;Sarcalumenin for the lumen of the longitudinal tubules of the sarcoplasmic reticulum;Ryanodine receptor Ca^2+^-release channel isoform RyR1 for the triad junction contact sites between transverse tubules and terminal cisternae of the sarcoplasmic reticulum;40S ribosomal protein SA for ribosomes;Vesicular transport factor for the Golgi apparatus;Lysosome-associated membrane glycoprotein 1 for lysosomes;Catalase for peroxisomes;Ubiquitin-conjugating enzyme E2 for proteasomes;Emerin for the myonucleus.

Mitochondria are essential for the extensive bioenergetic needs of multi-cellular systems, including highly specialized skeletal muscles [314] and have been extensively characterized by proteomics [315,316].

In skeletal muscles, excellent mitochondrial marker proteins [301,306,308] include:Voltage-dependent anion-selective channel protein VDAC1 for the mitochondrial outer membrane;Glutathione transferase for dynamic contact sites between the mitochondrial inner and outer membranes;Adenylate kinase isoform AK2 for the mitochondrial intermembrane space;NADH dehydrogenase for the mitochondrial inner membrane complex I;Succinate dehydrogenase for the mitochondrial inner membrane complex II;Cytochrome b-c complex for the mitochondrial inner membrane complex III;Cytochrome c oxidase for the mitochondrial inner membrane complex IV;ATP synthase for the mitochondrial inner membrane complex V;Isocitrate dehydrogenase for the mitochondrial matrix.

#### 3.3.4. Major Classes of Proteins in Skeletal Muscles

Major protein classes of skeletal muscles that are routinely detected by proteomics, as judged by bioinformatic PANTHER analysis [317], include DNA metabolism protein, RNA metabolism protein, calcium-binding protein, cell adhesion molecule, cell junction protein, chaperone, chromatin/chromatin-binding or -regulatory protein, cytoskeletal protein, defense/immunity protein, extracellular matrix protein, gene-specific transcriptional regulator, intercellular signal molecule, membrane traffic protein, metabolite interconversion enzyme, protein modifying enzyme, protein-binding activity modulator, scaffold/adaptor protein, storage protein, structural protein, transfer/carrier protein, translational protein, transmembrane signal receptor and transporter [290].

Crucial intracellular fractions of myofibers, the cytoskeletal network, metabolite transportation and the abundant extracellular matrisome of skeletal muscle tissues have been covered by proteomic analysis [25,26,27,28,290] and have confirmed the suitability of the following protein markers:Phosphofructokinase for the sarcosolic glycolytic pathway;Fatty acid-binding proteins for metabolite transportation in the cytosol;AlphaB-crystallin for the cellular stress response machinery;Desmin for intermediate filaments;Tubulins for microtubules;Myoglobin for intracellular oxygen transportation;Hemoglobin for external oxygen supply;Serum albumin for osmotic balancing in the extracellular space;Laminin-211 for the stabilization of the basal lamina;Collagens, such as isoform COL-VI, for the extracellular matrix;Periostin for the matricellular protein network;Decorin for the proteoglycan matrix.

#### 3.3.5. The Proteomic Profile of the Skeletal Muscle Secretome

Importantly, proteomics was applied to studying the muscle secretome, which can give crucial indications on the status of muscular changes in response to myogenesis, regeneration, disuse, neuromuscular disease, strenuous exercise, bioenergetic stress or aging [318,319,320,321].

Major muscle secretome components were identified as:Matrisomal proteins: perlecan, biglycan, decorin, alpha-dystroglycan, fibronectin, laminin, mimecan, nidogen, periostin, prolargin, matrix metalloproteinases and a large number of collagen isoforms;Cytokines and growth factors: angiopoietin, bone morphogenic protein, chemokines, cell adhesion molecules, complement factors, connective tissue growth factor, fibroblast growth factor, myostatin, insulin growth factor, transforming growth factor, tumor necrosis factor, and vascular endothelial growth factor;Essential enzymes: major glycolytic enzymes such as aldolase, glyceraldehyde-3-phosphate dehydrogenase, triosephosphate isomerase and pyruvate kinase, alcohol dehydrogenase, aldose reductase, mitochondrial ATP synthase, creatine kinase, superoxide dismutase, glycogen phosphorylase and protein disulfide isomerase;Enzyme inhibitors: cystatin, metalloproteinase inhibitors, various serpin isoforms and macroglobulin;Major contractile proteins: myosin heavy chains, myosin light chains, myosin-binding proteins, actin, titin (fragments), troponins and tropomyosins;Sarcomeric and non-sarcomeric cytoskeletal proteins: actin, F-actin capping proteins, alpha-actinin, cofilin, desmin, ezrin, filamin, myomesin, plectin, vimentin and vinculin.

Other muscle secretome proteins that are routinely detected by proteomics [321,322,323] include markers such as:Agrin for the neuromuscular junction;Annexins for the cellular repair mechanism;Heat shock proteins of the subclasses HspA, HspB and HspC, and endoplasmin, for the class of molecular chaperones involved in the cellular stress response;Calsequestrin, calmodulin and calreticulin for the Ca^2+^-handling apparatus;Carbonic anhydrase isoform CA3 for the regulation of carbon dioxide metabolism;Fatty acid-binding protein FABP3 for metabolite transportation.

In the context of biomarker research, the systematic cataloguing of the skeletal muscle secretome has established that many muscle proteins can be detected in suitable biofluids such as serum/plasma, urine or saliva for diagnostic purposes [324]. Muscle proteins that are associated with the intracellular fractions of myofibers can passively leak or be actively released into the systemic circulation due to strenuous exercise or traumatic muscle damage and then detected in biofluids. In addition to the above-listed proteomic markers CA3 and FABP3 [325], a variety of general muscle-derived serum markers are routinely assayed in single measurements or in combination. Muscle creatine kinase is the most frequently employed serum marker for the differential diagnosis of muscular abnormalities [45]. Other abundant and routinely used muscle damage markers include alanine amino transferase, aldolase, aspartate amino transferase, enolase, hydroxybutyrate dehydrogenase, lactate dehydrogenase, myosin light chain, troponin subunit TnI and myoglobin [326]. Although the 10 enzymes of glycolysis are present in almost all cell types in the body at high density, they nevertheless represent extremely abundant and distinct protein isoforms in the sarcosol of myofibers [309,327]. This makes specific muscle proteoforms of glycolytic enzymes, that can be routinely detected by advanced and high-throughput MS analysis, good candidates for establishing muscle-derived biomarkers of myofiber damage. This new array of markers for improved myofiber screening in health and disease exemplifies the vast impact proteomics has had on the cell biological characterization of skeletal muscle tissues over the last two decades of myology research.

#### 3.3.6. Progress of Cataloguing the Skeletal Muscle Proteome

A list of major publications that have focused on various aspects of skeletal muscle proteomics is provided in Table 1. These select papers include both primary research articles and key review articles outlining crucial findings in the field of basic and applied myology. The proteomic information summarizes the methodological approaches taken to establish the core protein components of skeletal muscles. The proteomic studies and reviews listed include (i) the initiation of the skeletal muscle proteome project [21,22], (ii) the systematic cataloguing of human skeletal muscles [20,23,26,27,37,53], (iii) proteomic surveys of fast versus slow human muscles [29,292], (iv) the analysis of human single myofibers [35,36,313], (v) the systematic cataloguing of animal skeletal muscles [25,28,288,290,328,329,330], (vi) the differential proteomic surveys of fast versus slow mouse muscles [30,31,32], (vii) profiling of animal single myofibers [25,33,277,311,312], (viii) the mass spectrometric analysis of muscle spindles [295], (ix) the profiling of the neuromuscular junction [296,297], (x) proteomic profiling of subcellular fractions (sarcolemma, sarcoplasmic reticulum, mitochondria, sarcosol, giant muscle protein assemblies, contractile apparatus) [92,140,175,304,305,306,308,309,310], (xi) the systematic cataloguing of the muscle secretome [318,319,321], (xii) profiling of the effects of exercise on the skeletal muscle proteome [38,39,40,289,331,332], (xiii) proteomic analysis of muscular disuse atrophy [261,333,334,335,336], (xiv) profiling of muscle plasticity [43,257] and (xv) the proteomic screening of skeletal muscle aging [34,130,135,264,291,337,338,339,340,341,342,343,344].

The coverage of the entire set of muscle-associated proteoforms is currently limited by the technically achievable enrichment and separation of dynamic protein populations prior to MS-based analysis. Future advances in single-cell preparations [12,166,188] and more efficient protein isolation during both top-down and bottom-up proteomics, in combination with further progress in the sensitivity of peptide/protein mass spectrometry [167,194,215,216,217,218,219,345], will be instrumental for the comprehensive cataloguing of the entire skeletal muscle proteome.

## 4. Conclusions

MS-based proteomics has been successfully applied to the comprehensive profiling of skeletal muscles. The usage of both top-down and bottom-up proteomic approaches has improved our cell biological knowledge of muscle protein diversity and the considerably heterogeneous composition of fast- versus slow-twitching fibers in skeletal muscles. Major classes of muscle-associated proteins that were identified by proteomics, and frequently exist as fiber type-specific isoforms, include components that are located in diverse cellular structures, including (i) thin actin filaments of the I-band, (ii) thick myosin filaments of the A-band, (iii) auxiliary titin and nebulin filaments, (iv) sarcomeric M-line complexes, (v) sarcomeric Z-disk complexes, (vi) the non-sarcomeric cytoskeleton, (vii) metabolic enzymes and transporters of anaerobic energy metabolism, (viii) metabolic enzymes and transporters of oxidative energy metabolism, (ix) cellular signaling complexes, (x) membranous, cytosolic and luminal ion-handling complexes, (xi) excitation-contraction coupling apparatus, (xii) molecular chaperoning system, (xiii) satellite cells and activated myogenic precursors, (xiv) myokines and fiber-derived factors of the muscle secretome, (xv) elaborate layers of the extracellular matrix consisting of the basal lamina, endomysium, perimysium and epimysium, (xvi) muscle-associated capillaries and (xvii) innervating motor neurons and insulating glial Schwann cells forming functional motor units. In addition, major organellar markers of skeletal muscles have been identified and characterized by proteomics, including proteins located in the nucleus, sarcoplasmic reticulum, Golgi apparatus, ribosomes, lysosomes, peroxisomes, proteasomes, transverse tubules, sarcolemma and mitochondria. This detailed proteomic knowledge can now be used to establish a systems biological view of skeletal muscle function.

## Figures and Tables

**Figure 1 cells-12-02560-f001:**
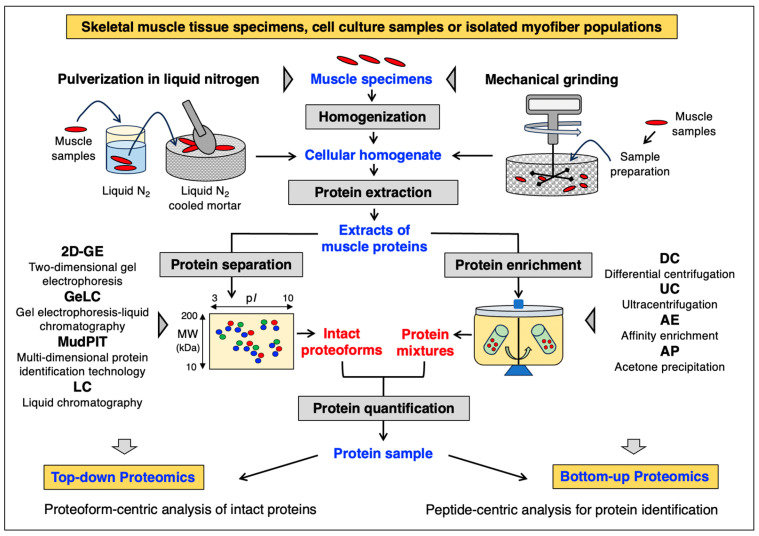
Overview of the initial preparative steps involved in top-down proteomics and bottom-up proteomics to study skeletal muscles. Abbreviations used: p*I*, isoelectric point; MW, molecular weight (in kDa).

**Figure 2 cells-12-02560-f002:**
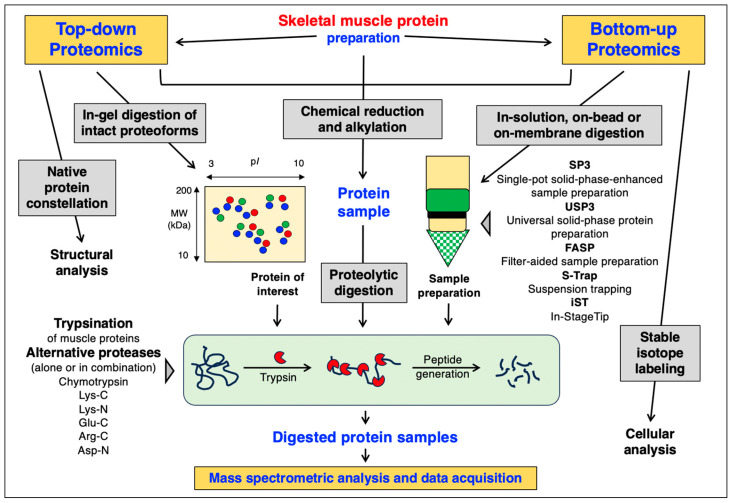
Comparison of the experimental steps involved in top-down proteomics versus bottom-up proteomics to study skeletal muscles. An overview of proteases used for the controlled protein digestion prior to peptide mass spectrometric analysis is provided. Abbreviations used: p*I*, isoelectric point; MW, molecular weight (in kDa).

**Figure 3 cells-12-02560-f003:**
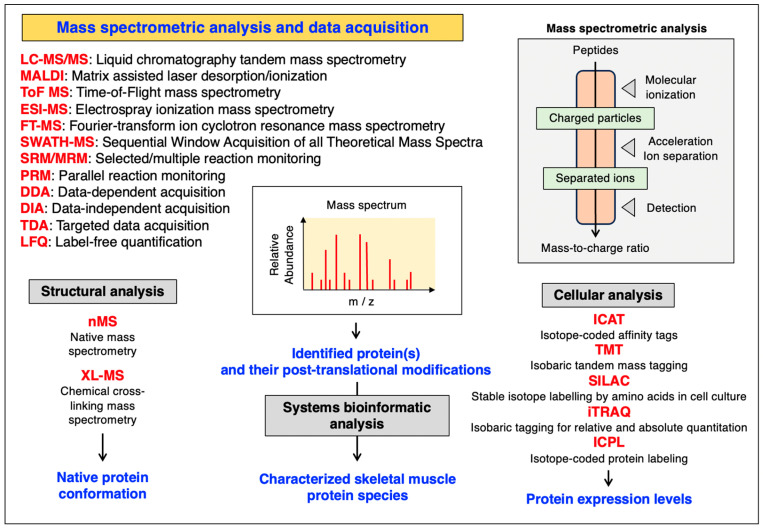
Overview of major mass spectrometric methods and data acquisition approaches that are routinely used in skeletal muscle proteomics (m/z, mass-to-charge ratio). The figure also lists key mass spectrometric methods employed for the structural analysis of native protein confirmation and the cellular analysis of protein expression levels.

**Figure 4 cells-12-02560-f004:**
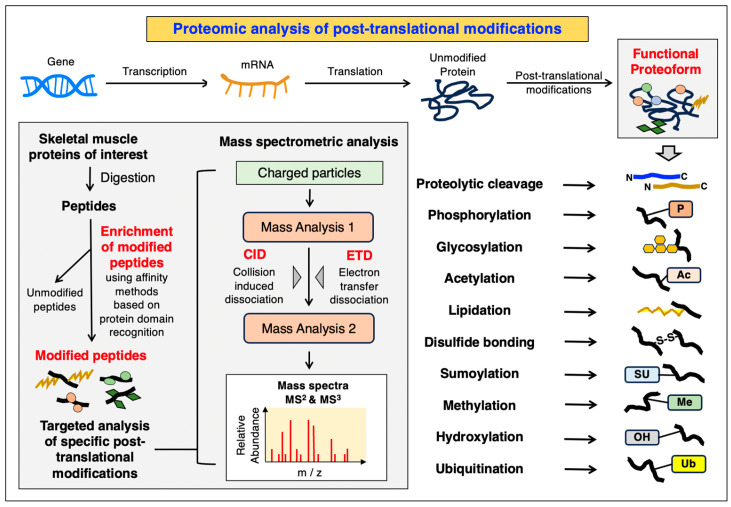
Overview of the complexity of post-translational modifications (PTMs) and how they are studied by mass spectrometric analysis (m/z, mass-to-charge ratio). Listed are major types of PTMs, including proteolytic cleavage, phosphorylation (P), glycosylation, acetylation (Ac), lipidation, disulfide bonding (S-S), sumoylation (SU), methylation (Me), hydroxylation (OH) and ubiquitination (Ub).

**Figure 5 cells-12-02560-f005:**
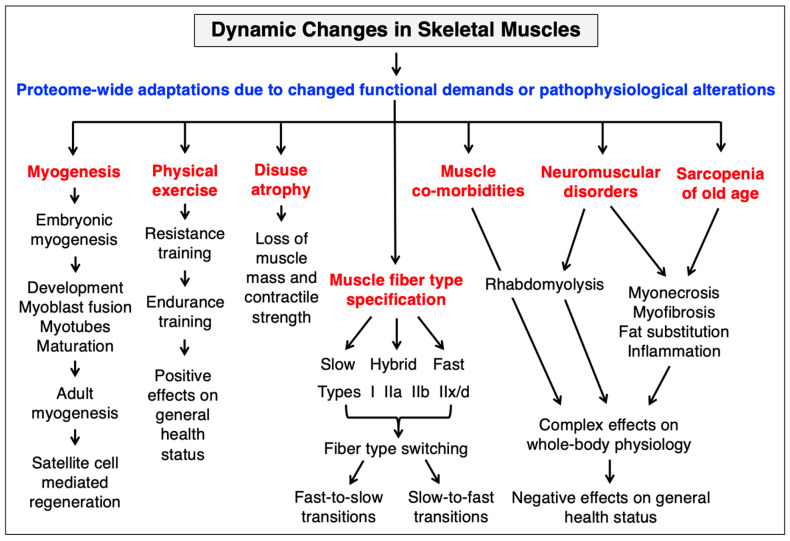
Overview of the extent of proteome-wide adaptations due to changed functional demands or pathophysiological alterations in the skeletal musculature. Major cell biological alterations occur during embryonic myogenesis, regenerative adult myogenesis, physical exercise, disuse atrophy, muscle fiber switching, muscular disorders and muscle aging.

**Figure 6 cells-12-02560-f006:**
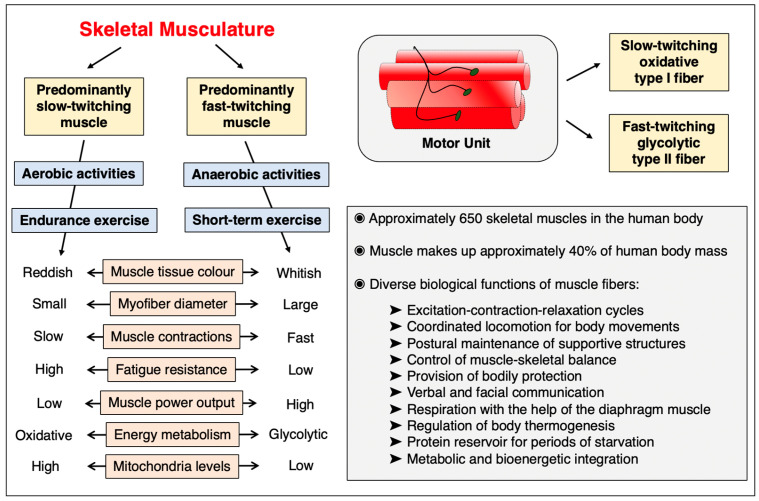
Summary of crucial cell biological characteristics of skeletal muscles. The figure includes a list of major histological, physiological and biochemical properties of the two main types of contractile fibers, i.e., slow-twitching type I myofibers and predominantly fast-twitching type II myofibers, as well as an inventory of the diverse functions of the skeletal musculature in the body.

**Figure 7 cells-12-02560-f007:**
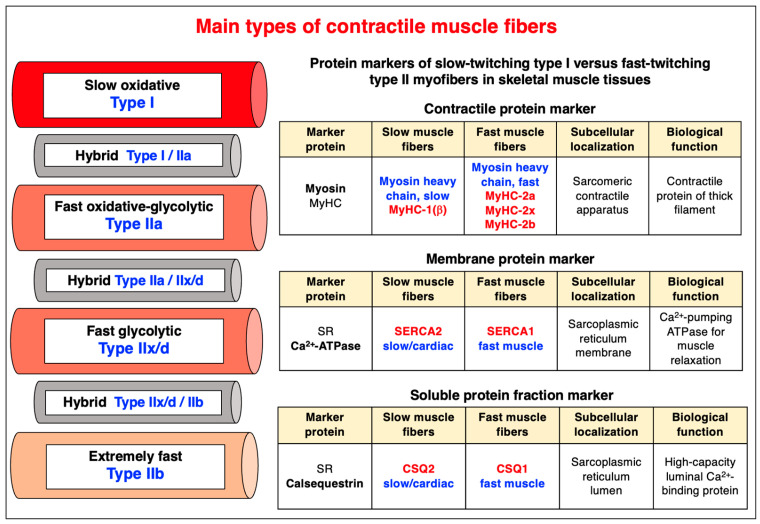
Overview of the main types of slow-twitching versus fast-twitching myofibers. The table at the right of the figure gives information on established marker proteins of fast versus slow myofibers. Abbreviations used: CSQ, calsequestrin; MyHC, myosin heavy chain; SERCA, sarcoplasmic or endoplasmic reticulum calcium ATPase; SR, sarcoplasmic reticulum.

**Figure 8 cells-12-02560-f008:**
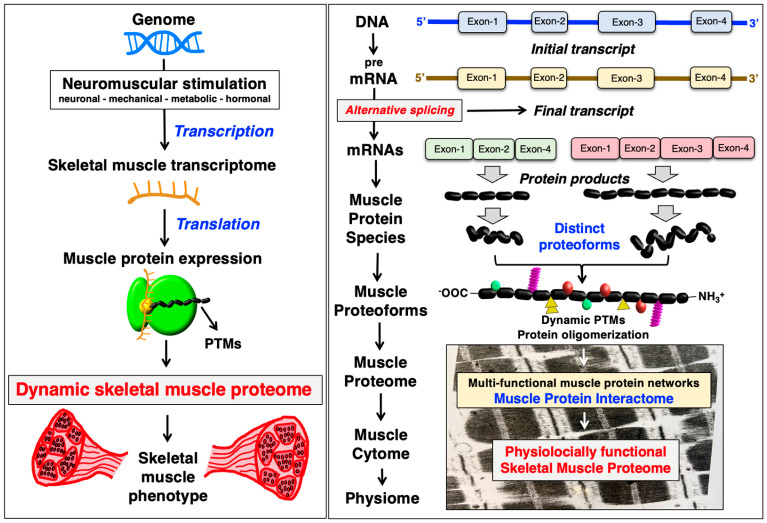
Diagrammatic presentation of the various layers of biological organization in skeletal muscle tissues. The figure summarizes the complex relationship and flow of information from gene to RNA/mRNA to protein/proteoform to the cytome and physiome in skeletal muscles. Proteomic diversity is due to alternative promoter usage, alternative post-transcriptional RNA splicing mechanisms and a large array of dynamic post-translational modifications (PTMs). For illustrative purposes, an electron microscopical image of rabbit skeletal muscle (*biceps femoris*) in the longitudinal direction is shown.

**Figure 9 cells-12-02560-f009:**
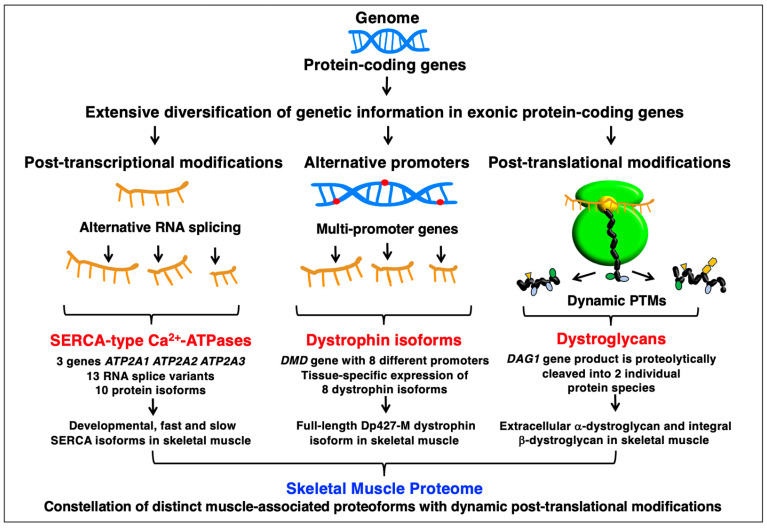
Proteomic complexity in skeletal muscles. The figure provides examples of the three main mechanisms that produce more than one specific proteoform per individual protein-coding gene, which are the underlying processes that produce a highly complex skeletal muscle proteome. As examples of post-transcriptional alternative RNA splicing, alternative promoter usage and post-translational modifications are shown the expression of the Ca^2+^-ATPase of the sarcoplasmic reticulum, the membrane cytoskeletal protein dystrophin and the sarcolemmal glycoprotein complex formed by the dystroglycans. Abbreviations used: DMD, Duchenne muscular dystrophy; PTMs, post-translational modifications; SERCA, sarcoplasmic or endoplasmic reticulum calcium ATPase.

**Figure 10 cells-12-02560-f010:**
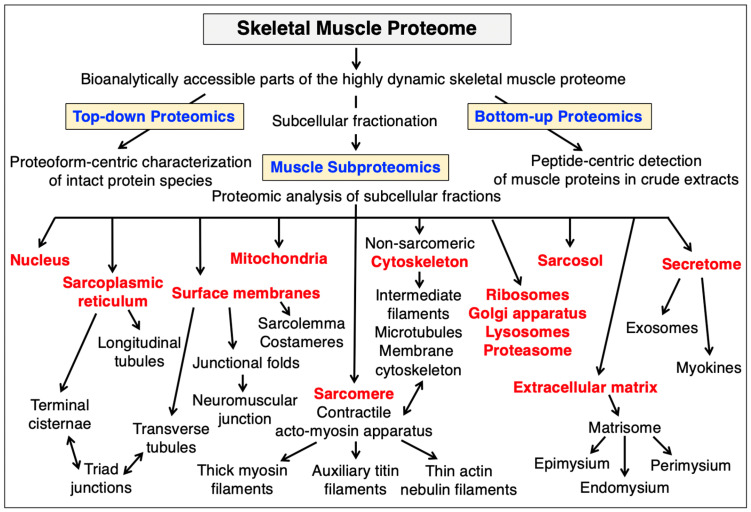
Overview of the bioanalytically accessible parts of the highly dynamic skeletal muscle proteome and the main subcellular structures of myofibers.

**Table 1 cells-12-02560-t001:** List of major mass spectrometric studies focusing on the cataloguing of the skeletal muscle proteome *.

Proteomic Analysis	Bioanalytical Focus	Technical Approach	References
Initiation of the skeletal muscle proteome project in the early 2000s; establishment of the mouse SWISS 2D-PAGE database	*Gastrocnemius* muscle, muscle tissues (*Mus musculus*, *Rattus norvegius*)	2D-GE, MALDI-ToF MS	Sanchez et al. [21]; Yan et al. [22]
Systematic cataloguing of human skeletal muscles	*Vastus lateralis* muscle, diagnostic biopsies (*Homo sapiens*)	1D-GE, LC-MS/MS	Højlund et al. [26]; Parker et al. [27]; Jiang et al. [53]
Differential proteomic surveys of fast versus slow human muscles	*Vastus lateralis*, *trapezius* and *deltoideus* muscles (*Homo sapiens*)	2D-GE, MALDI-ToF MS, laser capture micro- dissection	Capitanio et al. [29]; Stuart et al. [238]; Hadrévi et al. [292]
Profiling of human single myofibers	*Vastus lateralis* muscle (*Homo sapiens*)	Single-myofiber proteomics, LC-MS/MS	Murgia et al. [35]; Schiaffino et al. [36]; Momenzadeh et al. [313];
Description of the Human Skeletal Muscle Proteome Project	Overview of MS-based proteomics used to analyze human skeletal muscles	Literature search; summary of the human muscle proteome	Gonzalez-Freire et al. [20]; Gelfi et al. [23]; Capitanio et al. [37]
Systematic cataloguing of animal skeletal muscles	*Gastrocnemius* muscle, diaphragm muscle, C2C12 myoblast cell line (*Mus musculus*, *Rattus norvegius*, *Ovis aries*, *Sus scrofa*, *Bos taurus*)	2D-GE, TMT, FASP, LC-MS/MS	Deshmukh et al. [25]; Raddatz et al. [28]; Burniston et al. [288]; Murphy et al. [290]; Bouley et al. [328]; Xu et al. [329]; Wang et al. [330]
Differential proteomic surveys of fast versus slow mouse muscles	*Gastrocnemius*, *soleus*, *tibialis anterior* and *extensor digitorum longus* muscles (*Mus musculus*)	2D-GE, SILAC, LC-MS/MS, ESI-MS	Drexler et al. [30]; Gelfi et al. [31]; Okumura et al. [32]
Profiling of animal single myofibers	*Soleus*, *extensor digitorum* longus, *plantaris* and *vastus lateralis* muscles (*Mus musculus*, *Rattus norvegicus*)	Single-myofiber proteomics, ProFiT, FASP, LC-MS/MS, ESI-MS	Deshmukh et al. [25]; Eggers et al. [33]; Kallabis et al. [277]; Fomchenko et al. [311]; Melby et al. [312]
Profiling of muscle spindles	Spindles from *masseter* muscle (*Mus musculus*)	Dissection of muscle spindles, LC-MS/MS	Bornstein et al. [295]
Profiling of neuromuscular junction	Lower limb muscles, electric organ (*Homo sapiens*, *Torpedo californica*)	Dissection of neuromuscular junction region, TMT, LC-MS/MS	Jones et al. [296]; Mate et al. [297]
Profiling of myotendinous junction regions	*Gastrocnemius*, *soleus*, *extensor digitorium longus* and *tibialis anterior* muscles	Proteomic analysis of laser capture microscopy, LC-MS/MS	Can et al. [239]
Profiling of subcellular fractions (sarcolemma, sarcoplasmic reticulum, mitochondria, sarcosol, giant muscle protein assemblies, contractile apparatus)	Various skeletal muscles (*Mus musculus*, *Oryctolagus cuniculus*)	Enrichment and affinity isolation of subcellular fractions; LC-MS/MS	Murphy et al. [92]; Murphy and Ohlendieck [140]; Staunton and Ohlendieck [175]; Vitorino et al. [300]; Liu et al. [304,305]; Anunciado-Koza et al. [306]; Chae et al. [307]; Maughan et al. [309]; Murphy et al. [310]
Systematic cataloguing of the muscle secretome	*Quadriceps* muscle, primary human muscle cells, myoblasts (*Homo sapiens*)	1D-GE, 2D-GE, MALDI-ToF MS, ESI-MS, LS-MS/MS; literature review	Le Bihan et al. [318]; Hartwig et al. [319]; Florin et al. [321]
Profiling of the effects of exercise on the skeletal muscle proteome	*Vastus lateralis* and *soleus* muscles (*Homo sapiens*, *Rattus norvegius*)	2D-GE, LC-MS/MS, ESI-MS, PTM analysis; literature reviews	Cervone et al. [38]; Hesketh et al. [39]; Petriz et al. [40]; Hoffman et al. [248]; Hostrup et al. [249]; Koopman et al. [251]; Malik et al. [289]; Camera et al. [331]; Hesketh et al. [332]
Proteomic profiling of muscular disuse atrophy	*Gastrocnemius*, *soleus* and *vastus lateralis* muscles (*Homo sapiens*, *Rattus norvegius*)	2D-GE, single-fiber proteomics, LC-MS/MS	Doering et al. [258]; Li et al. [259]; Blottner et al. [260]; Murgia et al. [261]; Isfort et al. [333]; Moriggi et al. [334]; Sun et al., [335]; Lang et al. [336]
Proteomic profiling of muscle plasticity	*Tibialis anterior* muscle (*Oryctolagus cuniculus*)	Muscle electro-stimulation, 2D-DIGE, ESI-MS	Ohlendieck [43]; Donoghue et al. [257]
Proteomic profiling of skeletal muscle aging	*Vastus lateralis* and *gastrocnemius* muscles, single myofibers, freeze-dried muscle specimens (*Homo sapiens*, *Mus musculus*, *Rattus norvegius*)	2D-GE, 2D-DIGE, TMT, MALDI-ToF MS, ESI-MS, LC-MS/MS	Murgia et al. [34]; Gannon et al. [130]; O’Connell et al. [135]; Théron et al. [264]; Deshmukh et al. [291]; Gelfi et al. [337]; Doran et al. [338,339]; Staunton et al. [340]; Gueugneau et al. [341]; Ohlendieck [342]; Ebhardt et al. [343]; Ubaida-Mohien et al. [344]

* Abbreviations used: DIGE, difference gel electrophoresis; ESI, electrospray ionization; FASP, filter-aided sample preparation; GE, gel electrophoresis; LC-MS/MS, liquid chromatography tandem mass spectrometry; MALDI-ToF, matrix-assisted laser desorption/ionization time-of-flight; MS, mass spectrometry; ProFiT, proteomics high-throughput fiber typing; PTM, post-translational modification; SILAC, stable isotope labeling with amino acids in cell culture; TMT, isobaric tandem mass tagging.

## Data Availability

Not applicable.

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
