# Peer review of "Mass Spectrometry-Based Proteomic Technology and Its Application to Study Skeletal Muscle Cell Biology"

_cells, 2023, doi:10.3390/cells12212560_

Round 1
Reviewer 1 Report
Comments and Suggestions for Authors
This article offers a thorough and comprehensive overview of skeletal muscle proteomics and its significance in both basic and applied myology. The authors effectively explain the intricacies of studying the skeletal muscle proteome, emphasizing its dynamic nature and response to various factors such as exercise, disuse, and disease.
This is a well-written and informative review. The authors have demonstrated a deep understanding of the field and have clearly communicated the importance of skeletal muscle proteomics. However, the review could be even more effective in conveying its message to readers by introducing some changes. To enhance the overall readability of the review, it would be beneficial to address the following issues:
1-Wordiness and repetition: Some sentences and paragraphs are overly verbose and repetitive, which can make the reading experience tedious. Wordiness and repetition occur in certain sentences and paragraphs. By streamlining the language and avoiding unnecessary repetitions, the reading experience would be more enjoyable and efficient.
2-Prioritization of tables: Tables are a valuable tool for summarizing complex information and enhancing readability. However, the authors do not prioritize their use within the manuscript. Tables can summarize complex information and enhance readability. Integrating tables strategically throughout the review would facilitate comprehension and aid in quickly accessing important information
3-Scarcity of figures and diagrams: The figures and diagrams are well-designed and informative, but they are scarce compared to the long paragraphs. The authors should consider incorporating more visuals to break up the text and make the review more engaging and visually appealing.
it is recommended to be more concise and brief when explaining processes and findings of other studies and avoid the unnecessary use of long sentences.
I hope this feedback proves helpful in further improving the quality and impact of the review.
Author Response
Dear Reviewer 1,
Please find attached the revised R1-version of manuscript [cells-2675338] with the revised title ‘Mass spectrometry-based proteomic technology and its application to study skeletal muscle cell biology’.
We would like to thank you for your constructive criticism and positive evaluation of our manuscript.
As suggested, we have carried out revisions including changes to text, tables and figures, as outlined in detail in the responses to individual points. A separate manuscript version with changes highlighted in RED has been uploaded.
We hope that you will find the revised version of our article now acceptable for publication in Cells.
Best wishes
Kay Ohlendieck
Reviewer 1, Comment 1: ‘This article offers a thorough and comprehensive overview of skeletal muscle proteomics and its significance in both basic and applied myology. The authors effectively explain the intricacies of studying the skeletal muscle proteome, emphasizing its dynamic nature and response to various factors such as exercise, disuse, and disease. This is a well-written and informative review. The authors have demonstrated a deep understanding of the field and have clearly communicated the importance of skeletal muscle proteomics. However, the review could be even more effective in conveying its message to readers by introducing some changes.’.
Response: We would like to thank Reviewer 1 for the positive evaluation of our manuscript. As suggested, we have carried out revisions of text, tables and figures as outlined in detail in below responses to Reviewer 1.
Reviewer 1, Comment 2: ‘To enhance the overall readability of the review, it would be beneficial to address the following issues: 1-Wordiness and repetition: Some sentences and paragraphs are overly verbose and repetitive, which can make the reading experience tedious. Wordiness and repetition occur in certain sentences and paragraphs. By streamlining the language and avoiding unnecessary repetitions, the reading experience would be more enjoyable and efficient’.
Response: We have reorganized the manuscript and added additional figures and a summarizing table, as outlined in detail below. This makes the overall appearance of this article hopefully more accessible to the reader. We have revised the text accordingly, which should have removed repetitions. Additional sub-subheadings were introduced to better structure section 3.3. Listings in sentences are now numbered for better readability. Very long sentences were reorganized and this information is now displayed as bullet points. This should make it easier now for readers to extract information on proteomic markers of skeletal muscles. We have also introduced a list of abbreviations on revised Page 22.
Revised Page 15:
Established proteomic markers of adult fiber types [33-36] are:
• Myosin-7 (MyHC-I) for slow type I myofibers:
• Myosin-2 (MyHC-IIa) for fast type IIa myofibers:
• Myosin-4 (MyHC-IIb) for fast type IIb myofibers:
• Myosin-1 (MyHC-IIx/d) for fast type IIx/d myofibers:
Revised Pages 15-16:
Frequently used markers of the contractile apparatus and its auxiliary filaments within the sarcomeric structure [25-27,290] include:
• Myosin light chain MLC1/3 for the thick myosin-containing filament
• Alpha-actin for the thin actin-containing filament
• Tropomyosin isoform TPM2 for the tropomyosin complex
• Troponin subunit TnC for the troponin complex
• Titin for the sarcomere-spanning titin filament
• Myomesin for the sarcomeric M-band
• Alpha-actinin-2 for the Z-disk complex
Revised Pages 16-17:
Robust skeletal muscle markers that are closely associated with distinct subcellular structures [25,92, 304,305,309] include:
• Glycolytic enzymes for the sarcosol
• Dystroglycan for the sarcolemma
• Caveolin-1 for surface caveolae structures
• Dysferlin for the sarcolemmal repair apparatus
• Talin for costameres involved in lateral force transmission
• Laminin for the basal lamina
• Collagen for the extracellular matrix
• Dystrophin isoform Dp427-M for the sub-plasmalemmal membrane cytoskeleton
• Alpha-1S subunit of the L-type Ca2+- channel for the transverse tubules
• Fast SERCA1-type Ca2+-ATPases for the longitudinal tubules of the sarcoplasmic reticulum
• Calsequestrin for the terminal cisternae region of the sarcoplasmic reticulum
• Sarcalumenin for the lumen of the sarcoplasmic reticulum
• Ryanodine receptor Ca2+-release channel isoform RyR1 for the triad junction contact sites between transverse tubules and terminal cisternae of the sarcoplasmic reticulum
• 40S ribosomal protein SA for ribosomes,
• Vesicular transport factor for the Golgi apparatus,
• Lysosome-associated membrane glycoprotein 1 for lysosomes
• Catalase for peroxisomes
• Ubiquitin-conjugating enzyme E2 for proteasomes
• Emerin for the myonucleus
Revised Page 17:
In skeletal muscles, excellent mitochondrial marker proteins [301,306,308] include:
• Voltage-dependent anion-selective channel protein VDAC1 for the mitochondrial outer membrane
• Glutathione transferase for dynamic contact sites between the mitochondrial inner and outer membranes
• Adenylate kinase AK2 for the mitochondrial intermembrane space
• NADH dehydrogenase for the mitochondrial inner membrane complex I
• Succinate dehydrogenase for the mitochondrial inner membrane complex II,
• Cytochrome b-c complex for the mitochondrial inner membrane complex III
• Cytochrome c oxidase for the mitochondrial inner membrane complex IV
• ATP synthase for the mitochondrial inner membrane complex V
• Isocitrate dehydrogenase for the mitochondrial matrix
Revised Pages 17-18:
Crucial intracellular fractions of myofibers, the cytoskeletal network, metabolite transportation and the abundant extracellular matrisome of skeletal muscle tissues have been covered by proteomic analysis [25-28,290] and have confirmed the suitability of the following protein markers:
• Phosphofructokinase for the sarcosolic glycolytic pathway
• Fatty acid binding proteins for metabolite transportation in the cytosol
• AlphaB-crystallin for the cellular stress response machinery,
• Desmin for intermediate filaments
• Tubulins for microtubules
• Myoglobin for intracellular oxygen transportation
• Hemoglobin for external oxygen supply
• Serum albumin for osmotic balancing in the extracellular space
• Laminin-211 for the stabilization of the basal lamina
• Collagens, such as isoform COL-VI, for the extracellular matrix
• Periostin for the matricellular protein network
• Decorin for the proteoglycan matrix
Revised Page 18:
Major muscle secretome components were identified as:
• Matrisomal proteins: perlecan, biglycan, decorin, dystroglycan, fibronectin, lam-inin, mimecan, nidogen, periostin, prolargin, matrix metalloproteinases and a large number of collagen isoforms
• Cytokines and growth factors: angiopoietin, bone morphogenic protein, chemo-kines, cell adhesion molecules, complement factors, connective tissue growth factor, fibroblast growth factor, myostatin, insulin growth factor, transforming growth factor, tumor necrosis factor, and vascular endothelial growth factor
• Essential enzymes: major glycolytic enzymes such as aldolase, glyceralde-hyde-3-phosphate dehydrogenase, triosephosphate isomerase and pyruvate kinase, alcohol dehydrogenase, aldose reductase, mitochondrial ATP synthase, creatine kinase, superoxide dismutase, glycogen phosphorylase and protein disulfide iso-merase
• Enzyme inhibitors: cystatin, metalloproteinase inhibitors, various serpin isoforms and macroglobulin
• Major contractile proteins: myosin heavy chains, myosin light chains, myosin binding proteins, actin, titin, troponins, tropomyosins
• Sarcomeric and non-sarcomeric cytoskeletal proteins: actin, F-actin capping proteins, alpha-actinin, cofilin, desmin, ezrin, filamin, myomesin, plectin, vimentin, vinculin
Revised Page 18:
Other muscle secretome proteins that are routinely detected by proteomics [321-323] include markers such as:
• Agrin for the neuromuscular junction
• Annexins for the cellular repair mechanism
• Heat shock proteins of the subclasses HspA, HspB and HspC (and endoplasmin) for the class of molecular chaperones involved in the cellular stress response
• Calsequestrin, calmodulin, calreticulin for the Ca2+-handling apparatus.
• Carbonic anhydrase isoform CA3 for the regulation of carbon dioxide metabolism
• Fatty acid binding protein FABP3 for metabolite transportation
Reviewer 1, Comment 3: ‘2-Prioritization of tables: Tables are a valuable tool for summarizing complex information and enhancing readability. However, the authors do not prioritize their use within the manuscript. Tables can summarize complex information and enhance readability. Integrating tables strategically throughout the review would facilitate comprehension and aid in quickly accessing important information’.
Response: In response to this comment, a new Table 1 was introduced in the revised manuscript, which summarizes major mass spectrometric studies focusing on the cataloguing of the skeletal muscle proteome.
Revised Page 18: Section 3.3.6. (Progress of cataloguing the skeletal muscle proteome) now contains a new Table 1:
New text on Pages 18-19: ‘… A list of major publications that have focused on various aspects of skeletal muscle proteomics is provided in Table 1. These select papers include both primary research articles and key review articles outlining crucial findings in the field of basic and applied myology. The proteomic information summarizes the methodological approaches taken to establish the core protein components of skeletal muscles. The proteomic studies and reviews listed include the (i) initiation of the skeletal muscle proteome project [21,22], (ii) the systematic cataloguing of human skeletal muscles [20,23,26,27,37,53], (iii) proteomic surveys of fast versus slow human muscles [29,292], (iv) the analysis of human single myofibers [35,36,313], (v) the systematic cataloguing of animal skeletal muscles [25,28, 288, 290, 328-330], (vi) the differential proteomic surveys of fast versus slow mouse muscles [30-32], (vii) profiling of animal single myofibers [25,33,277,311,312], (viii) the mass spectrometric analysis of muscle spindles [295], (ix) the profiling of the neuro-muscular junction [296,297], (x) proteomic profiling of subcellular fractions (sarcolemma, sarcoplasmic reticulum, mitochondria, sarcosol, giant muscle protein assemblies, con-tractile apparatus) [92,140,175,304-306,308-310], (xi) the systematic cataloguing of the muscle secretome [318,319,321], (xii) profiling of the effects of exercise on the skeletal muscle proteome [38-40,289,331,332], (xiii) proteomic analysis of muscular disuse atrophy [261,333-336], (xiv) profiling of muscle plasticity [43,257] and (xv) the proteomic screening of skeletal muscle aging [34,130,135,264,291,337-344]. The coverage of the entire set of muscle-associated proteoforms is currently limited by the technically achievable enrichment and separation of dynamic protein populations prior to MS-based analysis. Future advances in single cell preparations [12,166,188] and more efficient protein isolation during both top-down and bottom-up proteomics, in combination with further progress in the sensitivity of peptide/protein mass spectrometry [167,194,215-219,345], will be instrumental for the comprehensive cataloguing of the entire skeletal muscle proteome.
Table 1. List of major mass spectrometric studies focusing on the cataloguing of the skeletal muscle proteome.
Please see the revised manuscript on Pages 19-21 for details on Table 1, which covers:
Initiation of the skeletal muscle proteome project
Systematic cataloguing of human skeletal muscles
Differential proteomic surveys of fast versus slow human muscles
Profiling of human single myofibers
Description of the Human Skeletal Muscle Proteome Project
Systematic cataloguing of animal skeletal muscles
Differential proteomic surveys of fast versus slow mouse muscles
Profiling of animal single myofibers
Profiling of muscle spindles
Profiling of neuromuscular junction
Profiling of myotendinous junction regions
Profiling of subcellular fractions
Systematic cataloguing of the muscle secretome
Profiling of the effects of exercise on the skeletal muscle proteome
Proteomic profiling of muscular disuse atrophy
Proteomic profiling of muscle plasticity
Proteomic profiling of skeletal muscle aging
Reviewer 1, Comment 4: ‘3-Scarcity of figures and diagrams: The figures and diagrams are well-designed and informative, but they are scarce compared to the long paragraphs. The authors should consider incorporating more visuals to break up the text and make the review more engaging and visually appealing’.
Response: To address this comment, figures were reorganized and new figures were added to the revised manuscript.
The revised manuscript now contains 10 figures:
Revised Page 4: Figure 1. Overview of the initial preparative steps involved in top-down proteomics and bot-tom-up proteomics to study skeletal muscles.
Revised Page 5: Figure 2. Comparison of the experimental steps involved in top-down proteomics versus bottom-up proteomics to study skeletal muscles and overview of proteases used for controlled protein digestion prior to peptide mass spectrometric analysis.
Revised Page 7: Figure 3. Overview of major mass spectrometric methods and data acquisition approaches that are routinely used in skeletal muscle proteomics. The figure also lists key mass spectrometric methods employed for the structural analysis of native protein confirmation and the cellular analysis of protein expression levels.
Revised Page 9: Figure 4. Overview of the complexity of post-translational modifications and how they are studied by mass spectrometric analysis.
Revised Page 10: Figure 5. Overview of the extent of proteome-wide adaptations due to changed functional demands or pathophysiological alterations in the skeletal musculature. Major cell biological alterations occur during embryonic myogenesis, regenerative adult myogenesis, physical exercise, disuse atrophy, muscle fiber switching, muscular disorders and muscle aging.
Revised Page 11: Figure 6. Summary of crucial cell biological characteristics of skeletal muscles. The figure includes a list of major histological, physiological and biochemical properties of the two main types of contractile fibers, i.e. slow-twitching type I myofibers and predominantly fast-twitching type II myofibers, as well as an inventory of the diverse functions of the skeletal musculature in the body.
Revised Page 12: Figure 7. Overview of the main types of slow-twitching versus fast-twitching myofibers. The table at the right of the figure gives information on established marker proteins of fast versus slow myofibers. Abbreviations used: CSQ, calsequestrin; MyHC, myosin heavy chain; SERCA, sarcoplasmic or endoplasmic reticulum calcium ATPase; SR, sarcoplasmic reticulum.
Revised Page 13: Figure 8. Diagrammatic presentation of the various layers of biological organization in skeletal muscle tissues. The figure summarizes the complex relationship and flow of information from gene to RNA/mRNA to protein/proteoform to the cytome and physiome in skeletal muscles. Proteomic diversity is due to alternative promoter usage, alternative post-transcriptional RNA splicing mechanisms and a large array of dynamic post-translational modifications (PTMs). For illustrative purposes, an electron microscopical image of rabbit skeletal muscle (biceps femoris) in the longitudinal direction is shown.
Revised Page 14: Figure 9. Proteomic complexity in skeletal muscles. The figure provides examples of the 3 main mechanisms that produce more than one specific proteoform per individual protein-coding gene, which are the underlying processes that produce a highly complex muscle proteome. As examples of post-transcriptional alternative RNA splicing, alternative promoter usage and post-translational modifications are shown the expression of the Ca2+-ATPase of the sarcoplasmic reticulum, the membrane cytoskeletal protein dystrophin and the sarcolemmal glycoprotein com-plex formed by the dystroglycans. Abbreviations used: DMD, Duchenne muscular dystrophy; PTMs, post-translational modifications; SERCA, sarcoplasmic or endoplasmic reticulum calcium ATPase.
Revised Page 15: Figure 10. Overview of the bioanalytically accessible parts of the highly dynamic skeletal muscle proteome and the main subcellular structures of myofibers.
Reviewer 1, Comment 4: ‘it is recommended to be more concise and brief when explaining processes and findings of other studies and avoid the unnecessary use of long sentences. I hope this feedback proves helpful in further improving the quality and impact of the review’.
Response: As already outlined in detail above, we have used numbering systems in long sentences that list various aspects of research findings, as well as used bullet points. This should give these sentences a better structure.
We would like to thank Reviewer 1 for the constructive criticism of our manuscript and hope that you will find the revised version of our article now acceptable for publication in Cells.
Reviewer 2 Report
Comments and Suggestions for Authors
This is an excellent review that clearly deserves publication. My comments for improvement are minor, but should be considered by the authors.
1) The title does not fully reflect the contents. Much of the manuscript focuses on the methodology of proteomics, which is applicable to multiple fields. I think people other than muscle biologists would benefit from reading the paper. Hence the authors might consider renaming the manuscript something like: Proteomic technology and its application to skeletal muscles
2) On page 5, “sample pre-treatment” should be labeled as (ii) instead of (iii)
3) On page 6, the following sentence is difficult to understand: “ToF-based analyzer determine efficiently the mass-to-charge ratio (m/z) using the time ions require to travel through an electric field under vacuum [189], combined with the MALDI-mediated detection of peptides”
4) Page 6, line 288: delete “to carry out”
5) Page 9, line 395 and Figure 3: constriction of blood vessels is under smooth muscle control and hence does not fit with this skeletal muscle review (perhaps true for constriction of some other organs as well).
6) Figure 2, legend: “muscle fiber shifting” is unclear. Perhaps use “muscle fiber switching” as in the figure.
7) Line 419 should not be indented, as I believe it is part of the previous paragraph.
8) Figure 3 legend: reference to the “table at the bottom left” should be “the bottom right”
9) Page 11 line 479: genes do not undergo alternative splicing, gene transcripts do
10) Page 12 line 510 and two other places in the text: “illustrative examples” seems redundant. Either illustrations or examples should be sufficient.
11) The authors might look again at section 3.3 to make sure it is logically organized. Is there a topic sentence for each paragraph that accurately represents the contents or do some contents belong in other paragraphs?
12) The paragraph beginning “Thus,” on page 15 line 647 is difficult to tie in with what preceded it. In particular, it is unclear what the initial “Thus,” is referring to.
Author Response
Dear Reviewer 2,
Please find attached the revised R1-version of manuscript [cells-2675338] with the revised title ‘Mass spectrometry-based proteomic technology and its application to study skeletal muscle cell biology’.
We would like to thank you for your constructive criticism and positive evaluation of our manuscript.
As suggested, we have carried out revisions including changes to text, tables and figures, as outlined in detail in the responses to individual points. A separate manuscript version with changes highlighted in RED has been uploaded.
We hope that you will find the revised version of our article now acceptable for publication in Cells.
Best wishes
Kay Ohlendieck
Reviewer 2, Comment 1: ‘This is an excellent review that clearly deserves publication. My comments for improvement are minor, but should be considered by the authors’.
Response: We would like to thank Reviewer 2 for the positive evaluation of our manuscript. As suggested, we have carried out revisions of text, tables and figures as outlined in detail in below responses.
Reviewer 2, Comment 2: ‘1) The title does not fully reflect the contents. Much of the manuscript focuses on the methodology of proteomics, which is applicable to multiple fields. I think people other than muscle biologists would benefit from reading the paper. Hence the authors might consider renaming the manuscript something like: Proteomic technology and its application to skeletal muscles’.
Response: We agree and have changed the title to: ‘Mass spectrometry-based proteomic technology and its application to study skeletal muscle cell biology’.
Reviewer 2, Comment 3: ‘2) On page 5, “sample pre-treatment” should be labeled as (ii) instead of (iii)’.
Response: We would like to thank Reviewer 2 for pointing out this mistake.
Revised Page 5: In the revised manuscript, it now correctly reads: ‘… cellular mixtures, (ii) sample pre-treatment, such as …’.
Reviewer 2, Comment 4: ‘3) On page 6, the following sentence is difficult to understand: “ToF-based analyzer determine efficiently the mass-to-charge ratio (m/z) using the time ions require to travel through an electric field under vacuum [189], combined with the MALDI-mediated detection of peptides”.’.
Response: To make this description clearer, we have split this sentence into two sentences, as follows:
Revised Page 6: ‘… ToF-based analyzer determine efficiently the mass-to-charge ratio (m/z) using the time ions require to travel through an electric field under vacuum [189]. This is combined with the MALDI-mediated detection of peptides [190].’.
Reviewer 2, Comment 5: ‘4) Page 6, line 288: delete “to carry out”.’.
Response: We have removed ‘to carry out’ from this sentence, which now correctly reads: ‘Micro-dissected tissue specimens can be used to generate cDNA libraries, the systematic genotyping of DNA, RNA transcript profiling, detailed protein pathway analyses, advanced biomarker discovery and spatial proteomics [233]’.
Reviewer 2, Comment 6: ‘5) Page 9, line 395 and Figure 3: constriction of blood vessels is under smooth muscle control and hence does not fit with this skeletal muscle review (perhaps true for constriction of some other organs as well)’.
Response: We agree and have removed this point from the text of the revised manuscript and the figure, which is now revised Figure 6.
Reviewer 2, Comment 7: ‘6) Figure 2, legend: “muscle fiber shifting” is unclear. Perhaps use “muscle fiber switching” as in the figure’.
Response: This has been corrected in the figure legend of the revised, which is now listed as Figure 5. It now reads ‘muscle fiber switching’.
Reviewer 2, Comment 8: ‘7) Line 419 should not be indented, as I believe it is part of the previous paragraph’.
Response: This has been corrected in the revised manuscript.
Reviewer 2, Comment 9: ‘8) Figure 3 legend: reference to the “table at the bottom left” should be “the bottom right”.;.
Response: This has been corrected. Since this figure has been changed, it now reads: ‘… The table at the right of the figure gives information on …’.
Reviewer 2, Comment 10: ‘9) Page 11 line 479: genes do not undergo alternative splicing, gene transcripts do’.
Response: We would like to thank Reviewer 2 to point out this confusing description. This has been corrected in the revised manuscript.
Revised Page 13: It now reads: ‘In humans, approximately 90% of transcripts of protein-coding genes have been estimated to undergo alternative splicing [278].’.
Reviewer 2, Comment 11: ‘10) Page 12 line 510 and two other places in the text: “illustrative examples” seems redundant. Either illustrations or examples should be sufficient’.
Response: This has been changed in the text and figure legend. It now reads as follows:
‘… of basic and applied myology. Examples of skeletal muscle proteomics are …’.
‘… of cellular biological processes. An example is the dystroglycan complex …’.
‘The figure provides examples of the …’.
Reviewer 2, Comment 12: ‘11) The authors might look again at section 3.3 to make sure it is logically organized. Is there a topic sentence for each paragraph that accurately represents the contents or do some contents belong in other paragraphs?’.
Response: To address this issue with Section 3.3., sub-subheadings have been introduced to give this section a better structure:
3.3.1. The status quo of the skeletal muscle proteome
3.3.2. The proteome of specialized cells and structures within skeletal muscles
3.3.3. The subproteome of skeletal muscles
3.3.4. Major classes of proteins in skeletal muscles
3.3.5. The proteomic profile of the skeletal muscle secretome
3.3.6. Progress of cataloguing the skeletal muscle proteome (including a new summarizing Table 1)
Reviewer 2, Comment 13: ‘12) The paragraph beginning “Thus,” on page 15 line 647 is difficult to tie in with what preceded it. In particular, it is unclear what the initial “Thus,” is referring to’.
Response: The beginning of this paragraph has been re-organized to avoid confusion. It now reads as follows: ‘In the context of biomarker research, the systematic cataloguing of the skeletal muscle secretome has established that many muscle proteins can be detected in suitable biofluids such as serum/plasma, urine or saliva for diagnostic purposes [324]. Muscle proteins that are associated with the intracellular fractions of myofibers can passively leak or being actively released into the systemic circulation due to strenuous exercise or traumatic muscle damage and then detected in biofluids. In addition to the …’.
We would like to thank Reviewer 2 for the constructive criticism of our manuscript and hope that you will find the revised version of our article now acceptable for publication in Cells.